# The Role of Non-Coding RNAs in ALS

**DOI:** 10.3390/genes16060623

**Published:** 2025-05-23

**Authors:** Alessandra Falduti, Adele Giovinazzo, Elisa Lo Feudo, Valentina Rocca, Filippo Brighina, Angela Messina, Francesca Luisa Conforti, Rodolfo Iuliano

**Affiliations:** 1Department of Experimental and Clinical Medicine, Campus S. Venuta, University Magna Graecia of Catanzaro, 88100 Catanzaro, Italy; alessandra.falduti@studenti.unicz.it (A.F.); elisa.lofeudo@studenti.unicz.it (E.L.F.); valentina.rocca@unicz.it (V.R.); 2Medical Genetics Unit, Renato Dulbecco University Hospital, 88100 Catanzaro, Italy; adele.giovinazzo@studenti.unicz.it; 3Neurology Unit, Department of Biomedicine, Neuroscience and Advanced Diagnostics (Bi.N.D.), University of Palermo, 90127 Palermo, Italy; filippo.brighina@unipa.it; 4Department of Biological, Geological and Environmental Sciences, University of Catania, Via S. Sofia 97, 95123 Catania, Italy; mess@unict.it; 5Medical Genetics Laboratory, Department of Pharmacy, Health and Nutritional Sciences, University of Calabria, 87036 Rende, Italy; francescaluisa.conforti@unical.it; 6Department of Health Sciences, Campus S. Venuta, University Magna Graecia of Catanzaro, 88100 Catanzaro, Italy

**Keywords:** amyotrophic lateral sclerosis (ALS), neurodegenerative disease, non-coding RNAs, microRNAs

## Abstract

Amyotrophic lateral sclerosis (ALS) is a progressive neurodegenerative disease that affects motor neurons, leading to muscle weakness, paralysis, and eventually death. The pathogenesis of ALS is influenced by genetic factors, environmental factors, and age-related dysfunctions. These factors, taken together, are responsible for sporadic cases of ALS, which account for approximately 85–90% of ALS cases, while familial ALS accounts for the remaining 10–15% of cases, usually with dominant traits. Despite advances in understanding and studying the disease, the cause of the onset of ALS remains unknown. Emerging evidence suggests that non-coding RNAs, including microRNAs (miRNAs), long non-coding RNAs (lncRNAs), and circular RNAs (circRNAs), play crucial roles in the pathogenesis of the disease. An abnormal expression of these molecules is implicated in various ALS-related processes, including motor neuron survival, protein aggregation, and inflammation. Here, we describe the dysregulation of non-coding RNAs in the pathogenic mechanism of ALS, highlighting the potential roles of miRNAs, lncRNAs, and circRNAs as biomarkers or therapeutic targets to examine the progression of the disease.

## 1. Introduction

Amyotrophic lateral sclerosis (ALS) is a fatal, neurodegenerative, and progressive disease that affects motor neurons in the brain, brainstem, and spinal cord, eventually leading to paralysis and death. The typical characteristics of the pathology are generalized muscle wasting (amyotrophy) and axonal loss in the lateral horns of the spinal cord (lateral sclerosis) [1,2].

In many patients, this disease has a spinal onset, characterized by weakness in the limbs; however, approximately one-third of patients have a bulbar onset of the disease, characterized by dysarthria (difficulties in speaking), dysphagia (difficulties in swallowing), and difficulties in chewing. The disease typically begins with focal muscle weakness, which gradually spreads to most muscles, including the diaphragm, ultimately leading to death from respiratory failure 3–5 years after diagnosis [3]. However, increasing evidence suggests that not only motor neurons but also other cell types are affected. In the spinal cord, cells such as astrocytes, microglia, oligodendrocytes, and interneurons are implicated in the degenerative process. Furthermore, serotonergic neurons in the brainstem and neurons in the frontal and temporal lobes have also been implicated [4]. Approximately 50% of patients may present extra-motor disorders in addition to the motor ones already mentioned; in particular, 10–15% of patients may develop frontotemporal dementia, while 35–40% may present cognitive and/or behavioral changes [5].

## 2. Epidemiology

ALS has an incidence of 1.75–3 per 100,000 people per year and a prevalence of 10–12 per 100,000 people in Europe, although there are important geographical differences [6,7].

However, despite its low incidence, ALS is not uncommon; indeed, the lifetime risk of developing the disease is approximately 1 in 400 [8].

Sexual dimorphism has also been observed in ALS; in fact, men have a greater propensity to develop the disease, with a male-to-female ratio between 1 and 3, depending on the population and geographical area [9]. Furthermore, the pathology tends to occur earlier in men than in women, although the survival time is similar in both sexes [10]. The phenotype also differs in the two sexes, with a prevalence of spinal-onset pathology in men and a prevalence of bulbar-onset pathology in women [10].

## 3. Etiology

Despite advances in understanding the roles of several factors involved in the pathology, the cause of the onset of ALS remains unknown. Similar to other neurodegenerative diseases, ALS may arise from a combination of genetic factors, environmental factors, and age-related dysfunctions. These combined factors are responsible for sporadic ALS, which accounts for approximately 85–90% of ALS cases, while familial ALS accounts for the remaining 10–15% of cases, usually with dominant traits [3]. Although associations have been identified between at least 40 genes and the disease, only 4 genes are considered the main ones, and they are responsible for approximately 48% of familial cases and approximately 5% of sporadic cases in the European population. These genes are *C9ORF72*, *SOD1*, *TARDBP*, and *FUS*, and they have provided valuable insights into the pathophysiology of ALS [1]. In ClinGen, the following 21 genes currently have a definitive classification as gene–disease validity for amyotrophic lateral sclerosis spectrum disorders: *ALS2*, *ANXA11*, *AR*, *ATXN2*, *C9ORF72*, *CHMP2B*, *FUS*, *GRN*, *KIF5A*, *NEK1*, *OPTN*, *PFN1*, *SLC52A2*, *SLC52A3*, *SOD1*, *SPG11*, *TARDBP*, *TBK1*, *UBQLN*, *VAPB*, and *VCP*.

However, the known genetic and environmental factors associated with the pathology seem not to be sufficient to cause and explain neurodegeneration and/or the presence of ALS in patients [11,12], as well as the phenotypic differences between the two sexes. An analysis of ALS phenotype networks has recently revealed an association between the disease and the response to radiation damage, a specific signature of women with bulbar-onset ALS [13]. Epigenetics may represent the bridge between genetic divergences, environmental factors, and the different observed phenotypes [14]. In this article, we explore what is known about the association between epigenetics, microRNAs, long non-coding RNAs, circular RNAs, and ALS.

## 4. Epigenetics and ALS

The term epigenetics, which literally means “above genetics”, refers to the study of alterations in gene expression that do not affect the genetic sequence and are both heritable and reversible. Epigenetic modifications normally occur during cell development and proliferation, but they can also occur as a result of random and/or environmental changes, thereby influencing cellular responses [15]. There are essentially three main epigenetic mechanisms: DNA methylation, histone modification, and non-coding RNA (ncRNA) modification [16]. In this article, we focus on the modifications of ncRNAs, particularly microRNAs and lncRNAs, and their potential implication in ALS.

## 5. MicroRNA and ALS

MicroRNAs (miRNAs) are a family of small, single-stranded non-coding RNAs (usually consisting of 20–22 nucleotides) that act as post-transcriptional regulators of gene expression in a sequence-specific manner. In fact, the 5′ regions of miRNAs bind and find complementarity with the 3′UTR regions of target messenger RNAs, causing their degradation or inhibition (Figure 1) [17]. Additionally, miRNAs can regulate other miRNAs and non-coding RNAs, such as lncRNAs and circRNAs, thereby enabling the control of cellular homeostasis [18]. 

miRNAs are generated from long primary transcripts called pri-miRNAs, which consist of a short double-stranded RNA region and a loop. The nuclear Drosha complex cleaves pri-miRNA to release an intermediate precursor known as pre-miRNA, which has a hairpin structure, and this is transported by Exportin-5 to the cytoplasm, where it is further cleaved by the Dicer complex to generate a mature miRNA. Finally, mature miRNAs are incorporated into the RNA-induced silencing complex (RISC), where they can hybridize with the 3′UTR region of their target mRNAs, either repressing translation or causing their degradation [19].

miRNAs are found in both intracellular and extracellular environments and are well conserved in various biological fluids, including plasma, serum, and cerebrospinal fluid. This allows us to measure their levels with far greater sensitivity and stability than other molecules, such as proteins [20,21]. Many studies conducted on biological samples, including blood and cerebrospinal fluid, have revealed the differential expression of miRNAs between healthy controls and patients with ALS, both familial and sporadic forms. This indicates the involvement of these small molecules in the pathogenesis of this neurodegenerative disease [22,23,24].

Thus, researchers are increasingly focusing on identifying miRNAs as potential biomarkers associated with this disease, with the aim of achieving faster and more accurate diagnoses, improved disease stratification, and monitoring [25]. In the following sections, the different roles of miRNAs in the pathogenesis of ALS are described. The principal genes and miRNAs involved in ALS pathogenesis and biological associations are summarized in Table 1.

## 6. miRNAs as Biomarkers and Therapeutic Targets

Currently, the diagnosis of ALS is predominantly clinical, as no definitive laboratory test is available to confirm the condition. Diagnostic procedures are based on neurological examination, electrophysiological testing, and the exclusion of other disorders with overlapping symptoms. The level of diagnostic certainty is classified according to the El Escorial criteria [37]. This reliance on clinical evaluation contributes to a diagnostic delay, with the average time from symptom onset to confirmed diagnosis being approximately 12 months [38]. This delay is particularly problematic given that existing treatments, riluzole and edaravone, the only FDA-approved drugs for ALS, offer only limited survival benefits [38]. Early intervention appears to be more effective, especially in the case of riluzole, which shows greater efficacy when administered in the initial stages of the disease [39]. These observations underline the need for early detection strategies to improve therapeutic impact. However, the development of early diagnostic tools is limited by the absence of validated molecular biomarkers capable of identifying ALS in presymptomatic stages or stratifying patients. Biomarkers Definitions Working Group has provided NIH biomarker definitions [40]. In ALS, reliable biomarkers could enable both earlier diagnosis and more targeted therapeutic approaches, potentially altering the course of the disease. In this context, miRNAs have emerged as promising candidates for biomarker development. Several studies have reported altered miRNA expression patterns in ALS patients [41,42,43,44,45,46,47,48], suggesting that specific profiles could serve as molecular signatures of the disease. These small non-coding RNAs exhibit tissue-specific expression and can be detected in various body fluids, including cerebrospinal fluid (CSF), blood, and urine, due to their encapsulation in exosomes or their binding to protein and lipoprotein complexes such as Argonaute proteins, which protect them from enzymatic degradation [49,50,51]. These features contribute to their stability and reproducibility, making them attractive as non-invasive peripheral biomarkers for ALS diagnosis [52]. Beyond their diagnostic value, miRNAs also hold therapeutic potential. Strategies aimed at suppressing upregulated miRNAs include the use of antagomirs and locked nucleic acids (LNAs) chemically modified oligonucleotides that mimic RNA structures and possess high binding affinity and nuclease resistance. 

These agents can inhibit miRNA activity by preventing interaction with target mRNAs and simultaneously promoting miRNA degradation [53]. On the contrary, in cases where miRNAs are found to be downregulated, therapeutic restoration can be achieved using synthetic miRNA mimics. These molecules reproduce the sequence and function of endogenous miRNAs, targeting the same mRNAs. Since reduced expression of specific miRNAs in ALS may result in the overproduction of pathogenic proteins, restoring their levels could offer protective effects [54]. However, the clinical application of miRNA mimics is limited by their short half-life and delivery challenges. To overcome these issues, viral vectors have been explored as delivery systems. However, efficient targeting of specific cell populations and traversal of the blood–brain barrier (BBB) remain major obstacles [55]. A significant breakthrough in this area is the identification of adeno-associated virus serotype 9 (AAV9), which is capable of crossing the BBB [56]. This discovery has renewed interest in gene therapy strategies for ALS, with recent studies exploring the potential of AAV-mediated RNA interference (RNAi) as a viable therapeutic approach [57].

## 7. miRNAs and Cytoplasmic Inclusions

Many miRNAs are involved in proper neuronal development and motor neuron survival, including miR-132-3p, making their correct metabolism and expression crucial.

Several proteins are involved in RNA maturation and metabolism, including TDP-43 and FUS, which are predominantly localized in the nucleus. TDP-43 is important for miRNA maturation because it promotes the interaction of pri-miRNAs with the nuclear Drosha complex, allowing their correct cleavage into pre-miRNAs. Additionally, cytoplasmic TDP-43 can also interact with the cytoplasmic Dicer complex, enabling the processing of pre-miRNAs [26]. FUS, however, facilitates miRNA maturation by activating the nuclear Drosha complex [58].

In ALS patients, both mutated and non-mutated TDP-43 and FUS proteins are predominantly found in the cytoplasm. Specifically, their presence has been detected in cytoplasmic inclusions (typical formations of neurodegenerative diseases) in affected cells, such as neurons and glial cells. This altered localization and lack of bioavailability of TDP-43 and FUS appear to alter their proper function, resulting in a reduced expression of miRNAs, including miR-132-3p, which are normally highly expressed in the neuronal cells of healthy individuals. These findings may contribute to understanding the pathogenesis of this neurodegenerative disease and to developing new therapeutic strategies for its treatment [26]. In 2011, Conforti F.L. et al. identified a new rare mutation, G376D in the *TARDBP* gene, in a woman from Southern Italy with ALS [59]. Family tree reconstruction allowed for an evaluation of plasma miRNA levels in asymptomatic and symptomatic family members carrying the G376D-*TARDBP* mutation. This study highlighted different expression levels of miR-132-5p, miR-132-3p, miR-124-3p, and miR-133a-3p, suggesting their potential role as ALS biomarkers associated with the G376D-*TARDBP* mutation [60].

## 8. miRNAs and Neuromuscular Junctions

In patients with ALS, some miRNAs are overexpressed, and this has been observed in studies on transgenic mice, such as those with the *SOD1* G93A mutation, who carry the mutated *SOD1* gene (one of the major susceptibility genes) with a glycine substitution at position 93. The expression of skeletal muscle miRNAs during ALS progression in symptomatic *SOD1* G93A transgenic mice was studied, and an elevated induction of miR-206 was found [27]. Under physiological conditions, miR-206 is highly expressed in skeletal muscle and is involved in maintaining neuromuscular synapses, regenerating neuromuscular junctions after injury, and regulating myoblast differentiation (Figure 2) [28,29]. In patients with ALS, miR-206 is overexpressed, and the reason for this is related to its ability to detect the damage and/or loss of motor neurons and its ability to promote the regeneration of synapses between individual neurons and muscles, thereby mitigating muscle damage. For this reason, the overexpression of this miRNA slows down the progression of the disease. To prove the importance of miR-206, the team bred miR-206−/− mice expressing a low copy number of *SOD1* G93A, and it was found that the loss of miR-206 accelerated disease progression and reduced survival by one month but did not affect disease onset; additionally, the initial progression of ALS was the same between *SOD1* G93A miRNA 206−/− transgenic mice and *SOD1* G93A mice [27,30]. Furthermore, previous works have provided evidence showing that miR-206 is critical for myoblast differentiation [61,62]; indeed, knockout mice for the skeletal muscle-specific Dicer-1 gene were found to have a reduced muscle mass due to hypoplasia [63].

Further studies have demonstrated that miR-206 inhibits the expression of histone deacetylase 4 (HDAC4), which is involved in the regulation of neuromuscular gene expression [64,65]. Specifically, miR-206 and HDAC4 have been shown to promote and prevent neuromuscular junction innervation, respectively, through opposite effects on fibroblast growth factor binding protein 1 (FGFBP1). MiR-206 activates FGFBP1, while HDAC4 inhibits it, and several studies have shown that the absence of FGFBP1 leads to the onset of abnormalities in the neuromuscular junction in *SOD1* G93A transgenic mice [66]. Previous studies have identified miR-206 as a reliable biomarker, and it is also believed that miR-206 may become a new therapeutic target for ALS due to its role in re-innervation and muscle repair [30,35].

Additionally, miR-23 has been shown to act as an inhibitor of peroxisome proliferator-activated receptor γ coactivator 1-α (*PGC-1α*), a transcriptional coactivator that plays a crucial role in regulating mitochondrial biogenesis and function, particularly in skeletal muscle in response to motor nerve activity [67]. Prior research has already confirmed that skeletal muscle mitochondrial dysfunction is involved in the severity and progression of ALS, and, as *PGC-1α* is implicated in mitochondrial biogenesis and function, the inhibition of this miRNA could be used to develop a therapeutic strategy to restore *PGC-1α* activity in ALS patients [67].

## 9. miRNAs and Neuroinflammation

It is now clear that neuroinflammation and the immune system play fundamental roles in the progression of ALS pathology through mechanisms such as the activation of microglia, the dysregulation of immune-related genes, and the recruitment of monocytes to affected tissues. miR-155, which has been shown to increase significantly in both sporadic and familial ALS, as well as in presymptomatic *SOD1* G93A transgenic mice [68,69,70], promotes an inflammatory state in tissues by activating Th17 cells and recruiting macrophages as part of the immune response [68,69,70]. Furthermore, miR-155, by binding and inhibiting the mRNA encoding the suppressor of cytokine signaling 1 (SOCS1), has been shown to promote the secretion of pro-inflammatory cytokines [31,32,71].

Butovsky et al. also demonstrated that, in *SOD1* G93A transgenic mice, miR-155 ablation reduced APOE expression in the spinal cord. APOE regulates lipid and cholesterol metabolism in the brain; moreover, in neurodegenerative diseases, it regulates the inflammatory response, although the mechanism remains unclear [33]. The fact that this miRNA reduced APOE expression in the spinal cord indicates that it played a crucial role as a pro-inflammatory mediator of microglia during the disease’s progression in these transgenic mice [70]. Anti-miR-155 has also been shown to promote longer survival in affected animals [69]. miR-125b has also been found to play an important role in the neuroinflammatory process. Some studies evaluating the expression profile of miRNAs in the microglia of *SOD1* G93A transgenic mice during the inflammatory process identified the involvement of miR-365 and miR-125b in pro-inflammatory signaling in microglia. Specifically, these miRNAs negatively regulated interleukin-6 (IL-6) and the signal transducer and activator of transcription 3 (STAT3), respectively, leading to an increase in tumor necrosis factor α (TNFα) expression and, consequently, the activation of pro-inflammatory signals [72].

The role of miR-125b in NF-kB signaling has also been investigated, and this miRNA has been shown to prolong NF-kB activation in microglia, having harmful effects on motor neurons. Therefore, the inhibition of miR-125b reduces NF-kB expression and protects motor neurons from death [73]. Considering all these findings, it can be asserted that there is a strong link between miRNAs and neuroinflammation, a process that contributes to the motor neuronal degeneration in neurodegenerative diseases, including ALS.

## 10. miRNAs, Neuronal Survival, and Apoptosis in ALS

By studying the expression profiles of miRNAs in *SOD1* G93A transgenic mice, Li et al. detected a reduced expression of miR-193b-3p, which is also observed in patients with ALS, both sporadic and familial forms. The team used motor neuron-like mouse hybrid cells (NSC-34 cells) transfected with miR-193b-3p to further investigate the role of this miRNA. The experiment demonstrated that the upregulation of miR-193b-3p leads to cell death in NSC-34 cells. Furthermore, miR-193b-3p was found to bind to and inhibit *TSC1* (tuberous sclerosis 1), an important regulator of cell differentiation and autophagy, in NSC-34 cells via the TSC1-mTOR pathway. Therefore, a reduction in miR-193b-3p expression contributes to the better regulation of autophagy and, consequently, greater neuronal survival [36].

Furthermore, the same team demonstrated that miR-183-5p regulates apoptotic and necrotic pathways at the neuronal level by targeting *RIPK3* and *PDCD4*. Through in cellulo experiments, they observed that the overexpression of this miRNA increases the survival of neuronal cells in stressful situations, while its depletion leads to neuronal death. This result confirms that cellular stress, as well as cell survival or death, is strongly regulated by miRNAs, which could represent new therapeutic targets for this pathology [74].

Another very important miRNA is miR-124, which is highly expressed in the central nervous system and is involved in processes such as neuritic growth, astrocyte differentiation, synaptic plasticity, and neurogenesis regulation. Additionally, miR-124 downregulates inflammation. The upregulation of this miRNA in *SOD1* G93A transgenic mice leads to neurodegeneration and microglial activation. To investigate the role of miR-124 in motor neuron degeneration, *SOD1* wild-type NSC-34 cells and NSC-34 cells expressing mutated *SOD1* G93A were used [34]. By transfecting both cell types with pre-miR-124, increased cell death through early apoptosis and reduced motor neuron activity in both cell types were reported, while the transfection of anti-miR-124 eliminated 100% of cell death in both types. Furthermore, normal levels of miR-124 are important for the growth and survival of motor neurons, as they can prevent damage from neurite outgrowth, inhibit microglial activation (which would otherwise lead to a pro-inflammatory state), and maintain a proper balance of mitochondrial dynamics. Thus, once again, these findings suggest the significant role of miRNAs as therapeutic targets in ALS pathology [34].

## 11. Long Non-Coding RNA and ALS

Long non-coding RNAs (lncRNA) are defined as a class of RNA molecules that are longer than 200 nucleotides and do not code for proteins. However, recent studies have identified some lncRNAs that encode polypeptides of <100 amino acids [75]. Despite this, these molecules play an important role in regulating gene expression and various cellular processes [76]. Research has shown that lncRNAs are involved in various biological functions, such as chromatin modification or transcriptional regulation, and post-transcriptional processing [77,78,79,80,81,82,83,84,85,86]. The role of these lncRNAs is fulfilled through direct or indirect interactions with genomic DNA, mRNA, and proteins [87]. Many studies have shown that NEAT1 (nuclear paraspeckle assembly transcript 1) promoters induce the expression of two types of non-coding RNAs (ncRNAs) in the human genome: NEAT1_1 lncRNA and NEAT1_2 lncRNA [88,89,90]. These isoforms overlap at the 5′ end while differing in length and the 3′ end; in fact, NEAT1_2 presents a short poly (A)-rich tract that presents similarity to metastasis-associated lung adenocarcinoma transcript 1 (MALAT1) [91]. Only the lncRNA NEAT1_2 is involved in the formation of a specific nuclear structure called “paraspeckles” (PSs) [92], and it plays a protective role in ALS progression through interactions with TDP-43 and FUS/TLS [93]. In detail, paraspeckles are composed of a core and a shell: the core contains the central region of NEAT1_2 and proteins such as NONO and SFPQ (splicing factor proline and glutamine rich) to form a heterodimer, while the shell contains the 3′ and 5′ ends of NEAT1_2 with other RBPs [94].

TDP-43 (TAR DNA-Binding protein-43) and FUS/TLS (fused in sarcoma/translocated in liposarcoma) are two RNA-binding proteins (RBPs) with nuclear localization predominantly involved in every step of RNA metabolism regulation. It is known from various investigations that mutations in TDP-43 mainly occur in the C terminus, containing the nuclear localization signal, and that they are responsible for the mislocalization of the nuclear protein in the cytoplasm of MNs, where it forms insoluble aggregates. At the same time, this may cause a loss of function of TDP-43 in the nucleus and a gain of toxic cytoplasmic function, both of which are detrimental to neuronal function and survival. As for TDP-43, FUS is predominantly a nuclear protein crucially involved in transcription, pre-mRNA splicing, and miRNA biogenesis [95]. However, it shuttles to the cytoplasm [96], particularly in neurons, indicating that it may participate in regulating mRNA transport into neurites and local protein translation in synapses [97]. Mutant FUS displays abnormal cytoplasmic localization in the neurons of ALS patients, where it accumulates in cytoplasmic inclusions, the stress granules (SGs).

Thus, the effect of these mutations in these genes is the formation of neuropathological inclusions, which are observed in 97% of patients with ALS [98]. In more detail, these stress granules (SGs) are membraneless cytoplasmic condensates that contain untranslated mRNA, pre-translational initiation factors, and RNA-binding proteins. The effect of the loss of nuclear localization and the gain of function in the cytoplasm is damage to various cellular processes in affected neurons, such as RNA processing, axonal transport, and neural transmission [99,100,101,102,103,104,105].

Under normal conditions, the expression of NEAT RNAs and paraspeckles is not observed in mature motor neurons. The formation of these SGs and cytoplasmatic aggregates is a reaction to events of excessive stress, for example, inflammation, viral infections, or oxygen deficiency. In the early phase of ALS, the expression of NEAT1_2 lncRNA increases, and the expression of NEAT1_1 RNA is downregulated through a decrease in alternative splicing. Under stressful conditions, the cells respond to this situation by sequestering TDP-43, FUS, and other splicing factors into cytoplasmic stress granules (SGs) to inhibit their accumulation. As the density and size of paraspeckles increase, they become less protective; thus, if stressful conditions continue when these structures are saturated, the protective function of paraspeckles collapses, and the expression of free NEAT1_1 increases. This increase in free NEAT1_1, decrease in NEAT1_2, and loss of protective function by paraspeckles cause the rapid acceleration of the cell cycle signaling that drives the cell to apoptosis, thus causing motor neuron death [93]. Another gene associated with ALS is *C9ORF72*, localized on chromosome 9 [106]. In particular, a GGGGCC (G4C2) repeat expansion in the first intron of the *C9ORF72* gene is a genetic cause of familial and sporadic ALS cases [107]. This expansion causes the production of sense and antisense RNA transcripts that lead, downstream of translation, to the production of five dipeptide repeat proteins (DPRs): poly-proline-alanine (PA), poly-glycine-alanine (GA), poly (GP), poly-glycine-arginine (GR), and poly-proline-arginine (PR) [97]. Although DPRs are hypothesized to be responsible for neurotoxicity, the molecular mechanisms through which they exert this effect are not yet fully understood. Studies have proven that poly-proline-arginine (poly-PR) is the most toxic DPR in vitro; in fact, it binds to and upregulates the expression of nuclear paraspeckle assembly transcript 1 (NEAT1), which is a scaffold RNA essential for paraspeckle formation. In addition, it is known that poly-PR interacts with some paraspeckle-localizing heterogeneous nuclear ribonucleoproteins (hnRNPs) and dysregulates their function. This demonstrates the neurotoxic action of polyPR. Furthermore, a significant association has been found between poly-PR and TDP-43: because of this interaction, the nuclear depletion of TDP-43 results in the upregulation of NEAT1 expression and, thus, neurotoxicity [108]. Gene therapy involves the use of antisense oligonucleotides (ASOs) to target the repeat expansion in the *C9ORF72* gene and to reduce RNA foci accumulation [109,110].

Another risk factor for ALS is the presence of CAG intermediate-length repeat expansions in the first exon of the *ATXN2* gene. ATXN2 repeat expansions were initially associated with spinocerebellar ataxia type 2 (SCA2); however, based on repeat length, they later became associated with other neurodegenerative diseases [111]. Ataxin-2 is a protein with ubiquitous cytoplasmic localization that regulates various cellular processes such as RNA processing, cellular growth, and stress granule dynamics [112]. This protein contains a polyglutamine (poliQ) repeat sequence in its N-terminal region, which has a length of 22–23 repeats in healthy individuals. When the number of expansions is between 27 and 34 repeats, this represents a major risk factor for the onset of ALS and is a cause of the production of abnormal protein ataxin-2; in fact, at the molecular level, poliQ expansion in ataxin-2 increases the interaction with TDP-43 and promotes the sequestration of TDP-43 to form irreversible cytoplasmic inclusions (SGs) with various RBPs (RNA-binding proteins) such as FUS, TDP-43, ataxin-2, and TAF15. Thus, this proves that ATXN2 is a potent modifier of TDP-43 toxicity [113]. The *ATXN2* locus is bidirectionally transcribed in ALS tissue: the antisense transcript ATXN2-AS with a CUG repeat expansion confers neurotoxic characteristics to cells, contributing to the increase in SCA2 and ALS pathogenesis. For this reason, ATXN2-AS could be a potential therapeutic target in SCA2 and ALS [114].

## 12. LncRNAs as Biomarker in ALS

The complexity of neurodegenerative diseases, such as Alzheimer’s disease, Parkinson’s disease, and amyotrophic lateral sclerosis, and the absence of effective targets, as well as highly sensitive biomarkers able to control disease progression, hinder the development of specific therapies [90]. Although many lncRNAs have been shown to be altered in numerous neurodegenerative diseases, studies exploring their potential use as biomarkers still remain limited. A study conducted in 2018 demonstrated the differential expression of lncRNAs in peripheral blood mononuclear cells (PBMCs) from five groups of subjects: sporadic ALS patients; FUS-, *TARDBP*-, and *SOD1*-mutated patients; and healthy controls [115]. They found a total of 293 differentially expressed lncRNAs (DE lncRNAs) in patients with sporadic ALS (sALS), of which 184 were antisense lncRNAs, and the others had unknown functions. In patients with FUS gene mutations, 21 deregulated lncRNAs were identified, of which 11 were antisense lncRNAs; in patients with *TARDBP* gene mutations, 7 types of antisense lncRNAs were detected; and, in *SOD1*-mutated patients, only 2 deregulated antisense lncRNAs were found. Another study investigated the deregulated expression of a panel of lncRNAs (linc-Enc1, linc–Brn1a, linc–Brn1b, linc-p21, Hottip, Tug1, Eldrr, and Fendrr) in a murine familial model of ALS (the *SOD1-G93A* mouse) in presymptomatic (8 weeks) and symptomatic (18 weeks) phases of the disease [116]. The results showed that there was a precise age-related and area-specific deregulation for each lncRNA. The lncRNA principally associated with ALS seemed to be linc-21, which was upregulated in the spinal cord and decreased in all areas of the central brain of the 18-week-old mice. The role of linc-21 is to suppress the p53 transcriptional pathway; thus, its upregulation could be the cause of the high cell death in the spinal cord. In this study, the expression of the human homologues of these lncRNAs was also investigated in an in vitro model, which included TUG1 (Tug1), TP53COR (linc-p21), HOTTIP (Hottip), PANTR1 (linc-Brn1a), ELDRR (Eldrr), and FENDRR (Fendrr). The results showed a trend of downregulation of all lncRNAs, especially HOTTIP and ELDRR [116]. This work can be considered a pilot study investigating the expression of lncRNAs in ALS patients with *SOD1* mutations. Moreover, the presence of NEAT1_2 lncRNA, the NEAT1_2 lncRNA/NEAT1_1 lncRNA expression ratio, and the formation of nuclear structures can be considered ALS-specific biomarkers because they reflect the progression of ALS. Thus, specific therapy could be developed to control the lncRNA expression levels in cerebrospinal fluid or serum [117].

## 13. LncRNAs as Potential Therapeutic Targets

Although many aspects still need to be investigated in the study of the role of lncRNAs in neurodegenerative diseases, the studies conducted to date present great potential for the future. The therapeutic strategies for ALS currently in use include drug therapy, gene therapy, immunotherapy, and stem cell–exosome therapy [118]. Drug treatment involves the use of compounds approved by the FDA, but the principal challenge to the discovery of new molecular therapies in the future is confirming the ability of drugs to cross the blood–brain barrier and reach the target site of action. Gene therapy involves the use of ASOs to silence the repeat expansion of GGGGCC of *C9ORF72* and target *ATXN2*; however, in addition to the direct targeting of ALS-associated genetic mutations, it is possible to intervene at the post-transcriptional or post-translational level modulating the expression or function of specific RNAs (sense or antisense) or proteins implicated in ALS. Furthermore, the pre-symptomatic administration of the polyGA vaccine was shown to reduce the number of inclusion bodies and prevent TDP-43 mislocalization, neuroinflammation, nerve axon damage, and motor deficits in a *C9ORF72* mouse model [119]. For the wide use of stem cell–exosome therapy, it is necessary to further analyze molecular mechanisms to improve our limited understanding of them. Based on the role of lncRNA1_2 in the formation of paraspeckles in ALS and on the neuronal protective function from degeneration by these paraspeckles, potential treatment strategies include the use of small compounds stimulating the NEAT lncRNA1_2 promoter and inhibiting the splicing factors that produce NEAT lncRNA1_1 [117]. However, it is very important to control the levels of free lncRNA1_1, which induces neuronal death; therefore, future targeted therapeutic strategies for ALS should aim to deplete this free lncRNA1_1 by using molecules that also preserve lncRNA1_2 levels. At present, it remains unclear whether specific oligos can be designed to perform this function and whether they could then function properly in vivo. Another target for future ALS treatments could be RNAs containing the G4C2 repeat expansions and DPRs derived from the *C9ORF72* gene: it has been found that the use of ASOs directed against these RNAs blocks the formation of ribonuclear foci, reducing their toxicity in motor neurons [117]. However, the results of using ASOs are limited because the foci of G4C2-RNA or DPRs are not the direct cause of the cell death that occurs during pathogenesis.

## 14. Circular RNAs and ASL

Circular RNAs are a class of closed RNA molecules (natural or synthetic), lacking 5′ and 3′ ends. For many years, they have been considered the result of protein-coding genes through abnormal/incorrect splicing, called “back-splicing”, of the precursor messenger RNA (pre-mRNA), during which a 5′ donor site connects with an acceptor site at the 3′ end, thus forming a 3′-5′ phosphodiester bond at the splice site and generating a circular RNA [120]. Instead, a small portion of these circular RNAs is derived from linear splicing, which leads to the formation of intronic circRNAs [107,121,122]. During the formation of intronic circRNAs, an upstream donor splice site binds to a downstream acceptor splice site to generate an intron lariat that evades debranching [120]. However, only recently has attention been paid to their role and natural biogenesis in various organisms.

Several studies have demonstrated how the presence of these circRNAs is ubiquitous and how, for this reason, they are involved in a variety of biological functions. For example, they can act as sponges by binding to miRNAs, thus freeing their targets; they can act as scaffold proteins by binding to other proteins in order to regulate cell proliferation and the immune response; or they can even act as promoters of protein expression and/or modulators of gene transcription [123]. The first study to define the link between circRNA and ALS was conducted by Errichelli et al., using motor neurons of mouse embryonic stem cells (ESCs). This study highlighted the role of FUS in binding to intronic sequences adjacent to circularizing exons in the process of circRNA formation. Therefore, it is known that FUS can have a positive or negative effect on the formation of circRNAs. Furthermore, it was hypothesized that the presence of pathogenic mutations in FUS could influence the biogenesis of circRNAs by compromising the regulation of splicing [107]. For example, D’Ambra et al. described an altered expression and localization of circ-Hdgfrp3 in murine motor neurons carrying ALS-associated FUS mutations. Through an imaging approach, they found that a proportion of circ-Hdgfrp3 was localized in neurites in both WT and mutant FUS motor neurons; however, under oxidative stress conditions, circ-Hdgfrp3 accumulated in stress granules (SGs) in WT motor neurons, and a higher proportion of circ-Hdgfrp3 was trapped in cytoplasmic aggregates in FUS-mutated motor neurons. After stress removal, the correct localization of circ-Hdgfrp3 was recovered, but a slower release of circ-Hdgfrp3 from FUS aggregates than from SGs was observed [122]. In another study, the role of RNA-binding proteins (RBPs) in the biogenesis of circRNAs was investigated through the creation of a mouse lacking TDP-43 in the forebrain. It emerged that this mouse had behaviors similar to those typical of mice with frontotemporal dementia (FTD). Through an analysis of RNA transcripts in the neocortex between mice lacking TDP-43 and control mice, a substantial difference in circRNA expression levels emerged; the biological significance of this, however, still requires examination [123]. Therefore, future studies on the structure and function of circRNAs could aid research examining the pathogenesis of neurodegenerative diseases to identify new treatment methods.

## 15. Conclusions

MicroRNAs (miRNAs) play a key role in the pathophysiology of ALS, a neurodegenerative disease characterized by motoneuron degeneration. Abnormal miRNA expression contributes to various pathogenetic mechanisms of ALS, promoting neuromuscular junction dysfunction, affecting neuroinflammation processes, and compromising neuronal survival. Other non-coding molecules, such as lncRNAs and circRNAs, contribute to this, highlighting the complexity of ALS pathogenetic mechanisms. Analyzing the role of miRNAs, lncRNAs, and circRNAs could help to identify new ALS biomarkers and new therapeutic opportunities with the aim of improving the prognosis of ALS patients.

## Figures and Tables

**Figure 1 genes-16-00623-f001:**
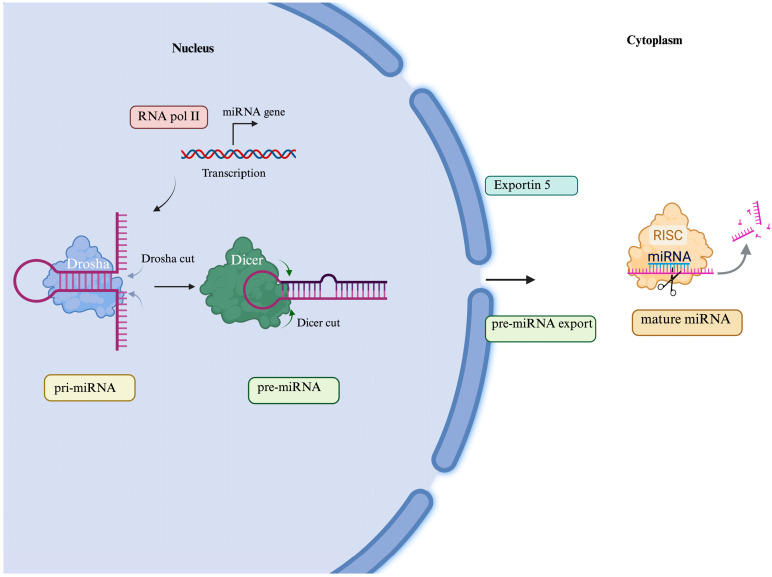
mi-RNA biogenesis process. miRNA biogenesis begins with the generation of the pri-miRNA transcript by RNA polymerase II. Pri-miRNAs are cleaved to produce the precursor-miRNA (pre-miRNAs) by the nuclear Drosha complex. Then, pre-miRNAs are exported into the cytoplasm by Exportin-5, where they are further cleaved by the Dicer complex to produce mature miRNA duplexes. Finally, mature miRNAs are incorporated into the RNA-induced silencing complex (RISC), where they can hybridize with the 3′UTR region of their target mRNAs, either repressing translation or causing their degradation [19]. Created in BioRender. www.biorender.com/ (accessed on 21 February 2025).

**Figure 2 genes-16-00623-f002:**
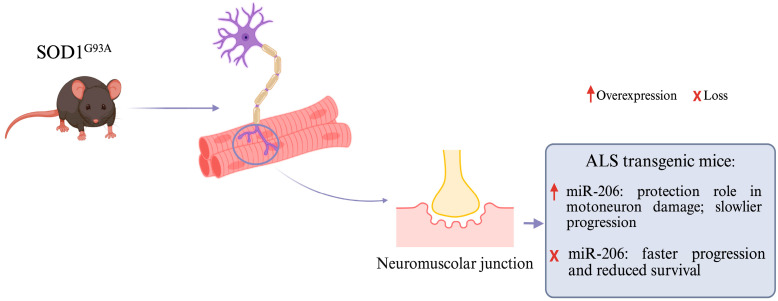
miRNAs involved in motoneuron function in ALS transgenic mice. miR-206 is one of the miRNAs overexpressed in ALS patients, and it performs a task protective against motoneuron damage, promoting a slower progression of disease. Studies about miR-206−/− transgenic mice with the *SOD1* G93A mutation prove that the loss of miR-206 accelerates disease progression and reduces survival by one month, even if it does not affect the disease onset between *SOD1 G93A* miRNA 206−/− transgenic mice and *SOD1 G93A* mice. Created in BioRender. www.biorender.com/ (accessed on 21 February 2025).

**Table 1 genes-16-00623-t001:** The principal genes and miRNAs involved in ALS pathogenesis and biological associations of ALS.

Gene	miRNAs	Involved in ALS Pathogenesis	Ref.
*TARDBP*	miR-132-5p miR-132-3p miR-124-3p miR-133a-3p	Formation of cytoplasmatic inclusion	[26]
*SOD1*	miR-206 miR-155 miR-125b miR-365 miR-193b-3p miR-124	Protective role in motoneuron damage; promote inflammatory response; increase of neurodegeneration	[27,28,29,30] [31,32,33,34]
*PGC-1α*	miR-23	Skeletal muscle mitochondrial dysfunction	[35]
*RIPK3*, *PDCD4*	miR-183-5p	Dysregulation of apoptotic and necrotic pathway of neuronal cells	[36]

## Data Availability

The original contributions presented in this study are included in the article. Further inquiries can be directed to the corresponding author.

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
