# Peer review of "The Role of Non-Coding RNAs in ALS"

_genes, 2025, doi:10.3390/genes16060623_

Round 1

Reviewer 1 Report

Comments and Suggestions for Authors

In this review “The Role of Noncoding RNAs in ALS”  by Falduti et al., described the dysregulation of ncRNA in pathogenic mechanism of ALS.

The title of study is very general. The title says the role of noncoding RNAs in ALS, but much emphasis was put on miRNAs. Very little on lncRNAs and circRNAs. If the authors do not want to emphasize the importance of other classes, it would be good to only focus on one class of RNA, miRNAs and discuss their implications in detail. There are already previous reviews covering the role of lncRNAs in ALS. This manuscript should only be considered if the authors made significant improvements and had answered all the concerns.

Further there are several major points to address, here below are some of those in detail:

  1. Throughout the manuscript, several sentences need to checked for clarity,
  2. The role of ncRNAs as biomarkers or therapeutic targets. Would be good to select either one of the aspects or try to discuss both aspects comprehensively. Only few studies have been referred (even not many recent ones). The authors do not discuss in detail or provide a good overview of recent developments in the field on why ncRNAs could be used as either biomarkers or therapeutic agents. Conclusions of each subheading are vague.
  3. Language needs to be checked. For e.g, Line 156-158. 173-176. This is throughout the paper, please read it carefully.
  4. Line 198-199; “Since previous studies have identified miR-206 as a reliable biomarker, it is believed that miR-206 may become a new therapeutic target for ALS”. It is confusing how a reliable biomarker could become a therapeutic target. Both are very different things. Please rewrite.
  5. Line 182-184: “Furthermore, Grifone et al. provided evidence for the importance of miR-206 during myogenesis; indeed, knockout mice for the skeletal muscle-specific Dicer-1 gene were found  to have reduced muscle mass due to hypoplasia [34]”. There is no evidence in this study about miR-206 or loss of Dicer on myogenesis. Please check the reference.
  6. Line 170-171: “It was found an elevated induction of miR-206, which is highly expressed”. This sentence should be rephrased.
  7. Figure2: The data in the figure is inferred from studies that were performed using HeLa, SY5Y and N2A cells (miR132-3p and 5p), not Motor neurons involved in ALS pathogenesis. Representing the data obtained from other cell lines and implying a similar mode of expression changes for ALS relevant celltypes is misleading. The authors should use studies that were performed using relevant cell types.
  8. Line 286-287: not clear what the sentence means. Should rephrase.
  9. Line314-320: Stress granules and paraspeckles are used interchangebily, where one is mainly cytoplasmic and other are nuclear structures. Further NEAT1 expression is mainly shown to be associated with paraspeckles. The authors should make this context clear and explain if NEAT1 is also found in SGs in the cytoplasm. It is not clear with the paragraph.
Comments on the Quality of English Language

English language should be checked. Several aspects in the manuscript are not conveyed well.

Author Response

In this review “The Role of Noncoding RNAs in ALS”  by Falduti et al., described the dysregulation of ncRNA in pathogenic mechanism of ALS.

The title of study is very general. The title says the role of noncoding RNAs in ALS, but much emphasis was put on miRNAs. Very little on lncRNAs and circRNAs. If the authors do not want to emphasize the importance of other classes, it would be good to only focus on one class of RNA, miRNAs and discuss their implications in detail. There are already previous reviews covering the role of lncRNAs in ALS. This manuscript should only be considered if the authors made significant improvements and had answered all the concerns.

RE: We would like to thank the Reviewer for his/her valuable comments that led us to improve the quality of our manuscript. In the new version of the manuscript, we have improved and extended the lnrcRNAs and circRNAs sections adding also some appropriate references.

Further there are several major points to address, here below are some of those in detail:

Throughout the manuscript, several sentences need to checked for clarity,

The role of ncRNAs as biomarkers or therapeutic targets. Would be good to select either one of the aspects or try to discuss both aspects comprehensively. Only few studies have been referred (even not many recent ones). The authors do not discuss in detail or provide a good overview of recent developments in the field on why ncRNAs could be used as either biomarkers or therapeutic agents. Conclusions of each subheading are vague.

RE: We would like to thank the reviewer for his/her useful comments. In this new version of the manuscript, we wrote a paragraph entitled “MiRNAs as biomarker and therapeutic target” and two other paragraphs entitled “LncRNAs as biomarkers in ALS” and “LncRNAs as potential therapeutic targets” adding recent references. We are confident that these additions can improve the quality of our manuscript.

Language needs to be checked. For e.g, Line 156-158. 173-176. This is throughout the paper, please read it carefully.

RE: We checked the language and the clarity of the manuscript as properly suggested by the Reviewer.

Line 198-199; “Since previous studies have identified miR-206 as a reliable biomarker, it is believed that miR-206 may become a new therapeutic target for ALS”. It is confusing how a reliable biomarker could become a therapeutic target. Both are very different things. Please rewrite.

RE: We rewrote this phrase.

Line 182-184: “Furthermore, Grifone et al. provided evidence for the importance of miR-206 during myogenesis; indeed, knockout mice for the skeletal muscle-specific Dicer-1 gene were found to have reduced muscle mass due to hypoplasia [34]”. There is no evidence in this study about miR-206 or loss of Dicer on myogenesis. Please check the reference.

RE: We rewrote this sentence.

Line 170-171: “It was found an elevated induction of miR-206, which is highly expressed”. This sentence should be rephrased.

RE: We rephrased the sentence.

Figure2: The data in the figure is inferred from studies that were performed using HeLa, SY5Y and N2A cells (miR132-3p and 5p), not Motor neurons involved in ALS pathogenesis. Representing the data obtained from other cell lines and implying a similar mode of expression changes for ALS relevant cell types is misleading. The authors should use studies that were performed using relevant cell types.

RE: In this new version of the manuscript, we modified the Figure 2 considering the reviewer's suggestions

Line 286-287: not clear what the sentence means. Should rephrase.

RE: We rephrased the sentence.

Line 314-320: Stress granules and paraspeckles are used interchangeably, where one is mainly cytoplasmic and other are nuclear structures. Further NEAT1 expression is mainly shown to be associated with paraspeckles. The authors should make this context clear and explain if NEAT1 is also found in SGs in the cytoplasm. It is not clear with the paragraph.

RE: The section containing the role of NEAT1 was completely rewritten according to the reviewer’s suggestions.

English language should be checked. Several aspects in the manuscript are not conveyed well.

RE: We carefully checked the English language as suggested.

Reviewer 2 Report

Comments and Suggestions for Authors

This is a minireview of the connections between several types of non-coding RNAs and ALS. The review is well-organized, informative and includes up-to-date information. It should prove  of value to anyone seeking a better understanding of this aspect of 
ALS pathology. Thus, acceptance for publication is recommended.

A suggestion for improvement:
Diagrams of miRNA function similar to Fig. 1 in its present form already appear in a number of textbooks and reviews. It is suggested to augment this figure by adding a depiction of ALS-specific dysfunction of this process.

Author Response

Reviewer 2

This is a minireview of the connections between several types of non-coding RNAs and ALS. The review is well-organized, informative and includes up-to-date information. It should prove  of value to anyone seeking a better understanding of this aspect of

ALS pathology. Thus, acceptance for publication is recommended.

RE: We would like to thank the Reviewer for her/his favourable comments.

A suggestion for improvement:

Diagrams of miRNA function similar to Fig. 1 in its present form already appear in a number of textbooks and reviews. It is suggested to augment this figure by adding a depiction of ALS-specific dysfunction of this process.

RE: We would like to thank the Reviewer for her/his suggestion. In this new version of the manuscript, we improved Figure 1 to focus on general miRNA biogenesis. We want to keep this figure to describe the overall process of miRNA biogenesis.

Round 2

Reviewer 1 Report

Comments and Suggestions for Authors

Review 2:

The authors have improved the review significantly by adapting the text and by expanding the information on miRNAs and lncRNAs.

Here are my suggestions:

  1. Line 31, Please change ASL – ALS.
  2. Check line 310 – 317: The information on miR-23 does not align well with the conclusion statement where the authors talk about miR-206? It reads as the conclusion statement was added wrongly here.
  3. Language and grammar still needs to be improved. For example sentences like in Line 487, “In the past, NEAT1 promoter was demonstrated to induce.”

What do the authors mean here? Why the sentence starts with “In the past”.?

  1. Line 490: “Differs to the length or differs in length”
  2. Line 650: No need to again abbreviate ALS every few paragraphs, as it was introduced in the beginning. Inconsistent with abbreviations. Please check this.
  3. Line 678: what does “step of ALS” mean? Please check the sentence.
  4. Line 733 -735: Please check for the naming of NEAT1 isoforms.
Comments on the Quality of English Language

Language and sentence structure still needs to improved. I have provided some examples in my suggestions.

Author Response

  1. Line 31, Please change ASL – ALS.

RE: We changed the acronym.

  1. Check line 310 – 317: The information on miR-23 does not align well with the conclusion statement where the authors talk about miR-206? It reads as the conclusion statement was added wrongly here.

RE: We would like to thank the reviewer for his/her useful comments. In this new version of the manuscript, we rewrote line 310 – 317 adding informations about miR-23 and reorganized paragraph. We are confident that these additions can improve the quality of our conclusion.

  1. Language and grammar still needs to be improved.

RE: We would like to thank the Reviewer for his/her valuable comments that led us to improve the quality of our manuscript. In the new version of the manuscript, we have carefully improved English language and grammar.

For example sentences like in Line 487, “In the past, NEAT1 promoter was demonstrated to induce.” What do the authors mean here? Why the sentence starts with “In the past”.?

RE: We rewrote this phrase.

  1. Line 490: “Differs to the length or differs in length”

RE: We rewrote this sentence.

  1. Line 650: No need to again abbreviate ALS every few paragraphs, as it was introduced in the beginning. Inconsistent with abbreviations. Please check this.

RE: We would like to thank the reviewer for his/her useful comment. We checked paragraphs and removed repeated abbreviations.

  1. Line 678: what does “step of ALS” mean? Please check the sentence.

RE: We rewrote the sentence.

  1. Line 733 -735: Please check for the naming of NEAT1 isoforms.

RE: We checked the naming of NEAT1 isoforms and its correct as reported by page 7 of article: Nishimoto Y, Nakagawa S, Okano H. NEAT1 lncRNA and amyotrophic lateral sclerosis. Neurochem Int. 2021 Nov;150:105175. doi: 10.1016/j.neuint.2021.105175. Epub 2021 Sep 2. PMID: 34481908.

In this new version our modification is highlighted in yellow.

Round 3

Reviewer 1 Report

Comments and Suggestions for Authors

Authors did a good job, they addressed all my concerns. I do not have any more questions.